# BRAF^V600E^ Expression in Thyrocytes Causes Recruitment of Immunosuppressive STABILIN-1 Macrophages

**DOI:** 10.3390/cancers14194687

**Published:** 2022-09-26

**Authors:** Catherine Spourquet, Ophélie Delcorte, Pascale Lemoine, Nicolas Dauguet, Axelle Loriot, Younes Achouri, Maija Hollmén, Sirpa Jalkanen, François Huaux, Sophie Lucas, Pierre Van Meerkeeck, Jeffrey A. Knauf, James A. Fagin, Chantal Dessy, Michel Mourad, Patrick Henriet, Donatienne Tyteca, Etienne Marbaix, Christophe E. Pierreux

**Affiliations:** 1CELL Unit, de Duve Institute, Université Catholique de Louvain, 1200 Brussels, Belgium; 2CYTF Platform, de Duve Institute, Université Catholique de Louvain, 1200 Brussels, Belgium; 3CBIO Unit, de Duve Institute, Université Catholique de Louvain, 1200 Brussels, Belgium; 4Transgenesis Platform, Université Catholique de Louvain, 1200 Brussels, Belgium; 5MediCity Research Laboratory and InFLAMES Flagship, University of Turku, 20500 Turku, Finland; 6LTAP Unit, IREC, Université Catholique de Louvain, 1200 Brussels, Belgium; 7GECE Unit, de Duve Institute, Université Catholique de Louvain, 1200 Brussels, Belgium; 8Walloon Excellence in Life Sciences and Biotechnology (WELBIO), 1300 Wavre, Belgium; 9Department of Otolaryngology Head & Neck Surgery in the Cleveland Clinic Lerner, College of Medicine of Case Western Reserve University, Cleveland, OH 44106, USA; 10Department of Medicine and Human Oncology & Pathogenesis Program, Memorial Sloan Kettering Cancer Center, New York, NY 10065, USA; 11FATH & MORF Unit, IREC, Université Catholique de Louvain, 1200 Brussels, Belgium; 12Surgery and Abdominal Transplantation Division, Cliniques Universitaires Saint-Luc, Université Catholique de Louvain, 1200 Brussels, Belgium

**Keywords:** *Stab1*^−/−^ mice, CD4/CD8 T cells, thyroid diseases

## Abstract

**Simple Summary:**

Incidence of thyroid cancer, including papillary thyroid cancer, is rapidly increasing. Oncogenes, such as the BRAF^V600E^, have been identified, and their effect on thyroid cancer cells have been studied in vitro and in mouse models. What is less understood is the impact of these mutations on thyroid cancer microenvironment and, in turn, the effect of changes in the microenvironment on tumor progression. We investigated the modifications in the cellular composition of thyroid cancer microenvironment using an inducible mouse model. We focused on a subpopulation of macrophages, expressing the STABILIN-1 protein, recruited in the thyroid tumor microenvironment following BRAF^V600E^ expression. CRISPR/Cas9 genetic inactivation of *Stablin-1* did not change macrophage recruitment but highlighted the immunosuppressive role of STABILIN-1-expressing macrophages. The identification of a similar subpopulation of STABILIN-1 macrophages in human thyroid diseases supports a conserved role for these macrophages and offers an opportunity for intervention.

**Abstract:**

Papillary thyroid carcinoma (PTC) is the most frequent histological subtype of thyroid cancers (TC), and BRAF^V600E^ genetic alteration is found in 60% of this endocrine cancer. This oncogene is associated with poor prognosis, resistance to radioiodine therapy, and tumor progression. Histological follow-up by anatomo-pathologists revealed that two-thirds of surgically-removed thyroids do not present malignant lesions. Thus, continued fundamental research into the molecular mechanisms of TC downstream of BRAF^V600E^ remains central to better understanding the clinical behavior of these tumors. To study PTC, we used a mouse model in which expression of BRAF^V600E^ was specifically switched on in thyrocytes by doxycycline administration. Upon daily intraperitoneal doxycycline injection, thyroid tissue rapidly acquired histological features mimicking human PTC. Transcriptomic analysis revealed major changes in immune signaling pathways upon BRAF^V600E^ induction. Multiplex immunofluorescence confirmed the abundant recruitment of macrophages, among which a population of LYVE-1+/CD206+/STABILIN-1+ was dramatically increased. By genetically inactivating the gene coding for the scavenger receptor STABILIN-1, we showed an increase of CD8+ T cells in this in situ BRAF^V600E^-dependent TC. Lastly, we demonstrated the presence of CD206+/STABILIN-1+ macrophages in human thyroid pathologies. Altogether, we revealed the recruitment of immunosuppressive STABILIN-1 macrophages in a PTC mouse model and the interest to further study this macrophage subpopulation in human thyroid tissues.

## 1. Introduction

In recent years, thyroid cancer (TC) has attracted more and more attention due to the rapid increase in its incidence. TC ranks fifth in incidence among all malignancies and is the most common endocrine tumor. In addition, the incidence rate is almost three times higher in women than in men [1]. The number of TC cases continues to increase not only due to external factors but mostly because of overdiagnosis of TC due to the higher sensitivity of diagnostic techniques [2]. Papillary thyroid carcinoma (PTC) accounts for 80–85% of TC cases [3], and about half of them carry the BRAF^V600E^ mutation [4,5]. The presence of this mutation is often associated with poor prognosis and distant metastasis. Although the 5 year survival rate is close to 98% when the cancer is diagnosed at an early stage, the survival rate drops to around 50% among people diagnosed with poorly differentiated thyroid cancer (PDTC) and/or distant-stage disease [6]. Another important challenge is the overdiagnosis of nodules or indolent microcarcinomas which leads to overtreatment and unnecessary surgeries [7,8]. It is, thus, important to better understand and characterize the biology and the microenvironment of TC to improve the diagnosis and treatment, especially when the cancer is poorly differentiated.

Solid tumors like TC are composed of various populations of cells including tumor cells, cancer stem cells, stromal cells, and immune cells. These cells, together with the extracellular matrix and the molecules produced, constitute the tumor microenvironment (TME), which plays a central role in tumor development and progression [9]. Indeed, in the TME, immune cells are supposed to recognize and eliminate cancer cells, but they rather progressively modify the TME to facilitate tumor progression. Tumor-associated macrophages (TAMs) are key components of the immune TME and are highly plastic cells with multiple functions [10]. TAMs are derived from tumor-infiltrating monocytes in peripheral blood or from expansion of tissue-resident macrophages [11]. TAM populations can transition between proinflammatory and anti-inflammatory states [12], and this polarization status depends on the pathological context and environment. Proinflammatory macrophages display antitumor effects such as the identification of tumor cells and their elimination. Anti-inflammatory/immunosuppressive macrophages are involved in tissue fibrosis and remodeling, angiogenesis, tumor progression, and immunoregulation toward immunosuppression [13].

The composition of the TC microenvironment depends on the mutations carried by the tumor cells, the cell type initiating the tumor, and the differentiation status of the tumor [14]. By considering the immune infiltration pattern of the TME and the gene expression profiling, a new classification of TC has been proposed. Two different clusters, the anaplastic thyroid cancer (ATC) and the PDTC, have been described, among which PTCs display a mixed immunological behavior with these two clusters [15]. Another study revealed that the presence of CD4+ T cells correlates with tumor size and aggressive features of TC, while the presence of cytotoxic CD8+ T cells in the TME is associated with a favorable prognosis in patients with PTC [16]. A reduced CD4+/CD8+ T-cell ratio is, thus, indicative of a good prognosis. TAM density also positively correlates with larger tumor size, lymph node metastasis, and decreased survival in PTC [17,18]. In a mouse model, PTC tumors expressing BRAF^V600E^ show high TAM infiltration in response to increased expression of TAM chemoattractant, colony stimulating factor-1 (CSF-1), and chemokine CCL2 by cancer cells [19]. In this mouse model and in TC in general, TAMs display an immunosuppressive phenotype [19,20]. Despite these important pieces of information on TC microenvironment, the complete identity of the immune microenvironment, as well as the complex and dynamic crosstalk between TC cells and their TME, still needs to be unraveled.

In this study, we aimed to better understand TME cellular composition, evolution, and role using a Tg-rtTA and tetO-BRAF^V600E^ double transgenic mouse model, mimicking BRAF^V600E^-dependent PTC (see [21] for details on the mouse model). Total RNA sequencing of thyroid tissue revealed major changes in different immune signaling pathways, as soon as 2 days after BRAF^V600E^ induction. Among the most deregulated genes, we found several chemokines and their receptors, which are mainly expressed by macrophages. Through immunohistofluorescence staining, we characterized these macrophages and identified a particular population which expressed LYVE-1 (a lymphatic vessel-specific glycoprotein), CD206, and STABILIN-1 (a scavenger receptor) markers. Using CRISPR/Cas9, we inactivated *Stabilin-1*, and found that its absence affected neither tumor size nor epithelial characteristics but was associated with a decrease in the intratumoral CD4+/CD8+ T-cell ratio, thereby supporting a potential immunosuppressive role for the STABILIN-1 macrophages. Lastly, we extended our observations to human tissues and showed the presence of these macrophages on sections from patients with benign and malignant thyroid pathologies.

## 2. Materials and Methods

### 2.1. Animals

Tg-rtTA and tetO-BRAF^V600E^ mice were obtained from J. A. Fagin (see [21] for details on the mouse model). Mice were intraperitoneally injected with a saline solution (CTL) or with doxycycline (1 μg/g body weight) every 24 h for a maximum of 4 days (Day 1 → Day 4) [22]. The *Stabilin-1* knockout mice were generated at the UCLouvain Transgenesis platform as described in [23]. Sequences targeted by the crRNAs (crRNA1: 5′–AGGAAAGAATTCAGAACTGG–3′; crRNA2: 5′–GTGATCGTTACCTATCCTCC–3′) were chosen to enclose the transcriptional initiation site and the entire first exon. Pups were screened for the deletion by classical genotyping PCR with the GoTaq R2 Hot Start Green Master Mix (Promega) with fw (5′–CCTCGGAAGCTGCCTAAGAT–3′), rv (5′–AAGGGAAAATGTACGGACACG–3′), fw’ (5′–TGTGGGGACAGTGATTGCAG–3′) and rv’ (5′–CCACCCACCACAAGCATAGA–3′) primers in the presence of DMSO 5%. Three heterozygous males (FVB mix B6D2) and four heterozygous females (FVB mix B6D2) were selected on the basis of Sanger sequencing of the PCR products and a ±1300 bp deletion to initiate the Tg-rtTA, tetO-BRAF^V600E^, *Stabilin-1* knockout colony. This colony was further backcrossed into FVB background. Mice were raised and treated according to the NIH Guide for Care and Use of Laboratory Animals, and experiments were approved by the University Animal Welfare Committee of the UCLouvain, Brussels, Belgium (2016/UCL/MD/005 and 2020/UCL/MD/011).

### 2.2. Tissue Collection

Mice were anaesthetized via injection with a xylazine (20 mg/kg)/ketamine (200 mg/kg) solution and sacrificed by cervical dislocation after blood puncture via the retro-orbital vein and cardiac flushing with PBS. Thyroid lobes were excised and either snap-frozen for RNA analysis or cut into 1 mm^3^ pieces for tissue dissociation and flow cytometry analysis. For histological or immunolabelling analysis on paraffin sections, thyroid lobes were extracted with the trachea, fixed in 4% paraformaldehyde at 4 °C overnight, and embedded in paraffin using a Tissue-Tek VIP-6 (Sakura). Sections of 6 µm were obtained with the microtome Micron HM355S (Thermo Scientific, Waltham, MA, USA). For immunohistofluorescence on frozen sections, thyroid lobes were collected and fixed in 4% paraformaldehyde at 4 °C overnight, followed by an overnight immersion in PBS/20% sucrose solution. Finally, tissues were embedded in PBS/15% sucrose/7.5% gelatin solution and stored at −80 °C. Sections of 7 µm were obtained with a cryostat (Thermo Scientific, Cryostar NX70). Frozen human tissue samples (sections or tissue pieces) were obtained from the Saint-Luc Hospital’s Biobank, Brussels. The study followed the Declaration of Helsinki and was approved by the local ethics committee of the UCLouvain, Brussels, Belgium (2017/25OCT/495 and 2017/04OCT/466).

### 2.3. Histology and Immunohistochemistry

After paraffin removal and rehydration, sections were either stained with hematoxylin and eosin for histological analysis or treated with 0.3% H_2_O_2_ for immunohistochemistry. Slides were processed as described [22] for antigen retrieval, permeabilization, blocking, and antibody staining (listed in Appendix A). For both histology and immunohistochemistry, slides were mounted in Dako aqueous Medium (Agilent Technologies, Santa Clara, CA, USA) and scanned with the panoramic P250 digital slide scanner (3DHistech, Budapest, Hungary).

### 2.4. Immunohistofluorescence and Image Analysis

For LYVE-1, CD11b, CD206, and STABILIN-1 labeling, frozen sections were used. Tissues were treated as described in [23] for permeabilization, blocking, and staining steps. For F4/80, LAMP-1, and TG staining, paraffin sections were used. After paraffin removal and rehydration, sections were treated as for immunohistochemistry (see above) for antigen retrieval, permeabilization, blocking, and labeling with primary antibodies (listed in Appendix A). For CD3 and CD8 labeling, tissues on paraffin sections were treated as described in [24]. Sections were then blocked with PBS/0.3% triton X-100/10% BSA/3% milk solution for 45 min, and then incubated with anti-E-CADHERIN antibody overnight at 4 °C. Secondary antibodies coupled to Alexa-488, -568, or -647 (Invitrogen, Carlsbad, CA，USA) were used, as well as a fluorescent nuclear dye (Sigma, Hoechst 33258, Burlington, MA, USA). Slides were finally mounted with Dako Fluorescent Mounting Medium, and then scanned with the Pannoramic P250 Digital Slide Scanner (3DHistech) or observed with the Zeiss Cell Observer Spinning Disk (COSD) confocal microscope. Images were analyzed with Zen software (Zeiss, Oberkochen, Germany). Image quantifications of CD3 and CD8 cells were performed with the HALO software and its Cytonuclear FL module (Indicalabs, Albuquerque, NM, USA). Due to the lack of a validated anti-CD4 antibody on tissue sections, CD4+ T cells were considered as CD3+ CD8− cells and CD8+ T cells were considered as CD3+ CD8+ cells.

### 2.5. Blood Analysis and ELISA

Blood was collected with sodium-heparinized capillary tubes from the retro-orbital sinus. Blood was analyzed using the MS9-5s hematology analyzer (MS Laboratoires, Osny, France). For ELISA, blood was centrifuged at 1880× *g* for 10 min to collect the plasma fraction. Plasma was then centrifuged at 2500× *g* during 10 min to eliminate remaining red blood cells and platelets. Plasma TNFα was quantified using the mouse TNFα uncoated ELISA kit (Invitrogen) according to the manufacturer’s instructions.

### 2.6. mRNA Quantification

Total RNA was extracted from mice thyroid tissues using TRIzol Reagent (Thermo Scientific, Oxfordshire, UK), as described in [25], followed by an additional phenol/chloroform purification step. Total RNA was extracted from human thyroid tissues using the miRNeasy Mini kit (Qiagen, Venlo, The Netherlands) according to the manufacturer’s instructions and as described in [26]. Tissue RNA concentration was measured by spectrophotometry (Thermo Scientific, NanoDrop 8000). Briefly, 500 ng of total RNA from each sample was reverse-transcribed using M-MLV Reverse Transcriptase (Invitrogen) and random hexamers according to the manufacturer’s instructions. Reverse transcription quantitative PCR (RT-qPCR) was performed on 15 ng of cDNA, as described [27,28]. Primer sequences are listed in Appendix A. Data were analyzed using the ΔΔCT method, with the geometric mean of *Gapdh* and *Rpl27* expression as reference genes for mice tissues, and *RPL13A* expression as the reference gene for human tissues.

### 2.7. mRNA Sequencing and Bioinformatic Analysis

DNase-treated RNA samples from control (CTL), Dox 2 days (Day 2), and Dox 4 days (Day 4) thyroid (*n* = 4; two males and two females for each group) were sequenced by GENEWIZ-NGS Europe (Germany) using the Illumina HiSeq platform, 2 × 150 bp PE configuration. The RNA library was prepared with polyA selection. Read quality control was performed using FastQC software v0.11.8 [29]. Low-quality reads were trimmed, and adapters were removed using Trimmomatic software v0.38 [30]. Reads were aligned using HISAT2 software v2.1.0 [31] on GRCm38genome. Gene counts were generated using htseq-count from HTseq package v0.11.2 [32] and the Ensembl Mus_musculus.GRCm38.94.gtf annotation file. All programs were run with default parameters. Differential expression analyses were performed using DESeq2 v1.22 [33], on R version 3.5.1. Genes with an adjusted *p*-value <0.05 (Benjamini–Hochberg method for multiple-testing correction) and an absolute log_2_ fold change >2 were considered as differentially expressed. Enrichment pathway analyses were performed with MouseMine software, using standard parameters (URL: https://www.mousemine.org/mousemine/begin.do, accessed on 25 September 2022).

### 2.8. Flow Cytometry

The spleen and thymus were crushed with the back of a 5 mL syringe in 5 mL of IMDM medium (Gibco, 21980-032, Waltham, MA, USA) supplemented with 50 µM β-mercaptoethanol (Gibco, 31350-010), GlutaMAX^TM^-1 (Gibco, 35050-038), and 5% FBS. Cell suspension was filtrated on a 70 µm filter and centrifuged at 300× *g* for 6 min. The splenic and thymic cell suspension, as well as blood samples, was resuspended in 4 mL of Red Blood Cell lysis buffer (eBioscience, 00-4300-54, San Diego, CA, USA) for 5 min at RT. After stopping the reaction with IMDM media, cells were passed through a 40 µm filter, centrifuged at 300× *g* for 6 min, and finally resuspend in PBS/1 mM EDTA/1% FBS solution before counting. To identify the immune populations, 1 × 10^6^ cells were blocked in PBS/1 mM EDTA/1% FBS/1 µg of TrueStain FcX^TM^ (Biolegend, 101320, San Diego, CA, USA).

Thyroid lobes (eight lobes/condition/experiment) were cut into 1 mm^3^ pieces and processed into single-cell suspensions with the Mouse Tumor Dissociation Kit (Miltenyi Biotec, 130-096-730, Seoul, Korea) diluted in Ca^2+^-free DMEM (Gibco, 21068-028) for 40 min at 37 °C with up and down mixing using a P1000 every 8 min. Then, the cell suspension was passed through a 40 µm filter and diluted in Ca^2+^-free DMEM/1 mM EDTA pH8/20% FBS to stop dissociation. Cells were centrifuged and resuspended in the same blocking solution as for splenic, thymic, and blood cells for 10 min.

All cell samples were stained with anti-CD45, anti-CD11b, anti-CD3, anti-CD4, anti-CD8a, anti-CD19, and anti-CD49b fluorochrome-conjugated antibodies. Dead cells were visualized using DAPI (Fisher Scientific, D3571, Waltham, MA, USA) staining. Data were acquired on the BD LSRFortessa^TM^ Cell Analyzer and analyzed with FlowJo^TM^ software (BD Biosciences, East Rutherford, NJ, USA). All antibodies and working conditions are recapitulated in Appendix A.

### 2.9. Western Blotting

Western blotting was performed as described [34]. Briefly, lymph nodes from Stabilin-1 WT (*Stab1*^+/+^) and Stabilin-1 KO (*Stab1*^−/−^) mice were lysed in RIPA buffer, and the protein concentration was measured using the bicinchoninic acid assay. Then, 50 µg of protein was loaded and electrophoresed through a 4–15% Mini-PROTEAN TGX precast gel (BIO-RAD, 4561084, Hercules, CA, USA). Proteins were transferred onto a PVDF membrane (0.45 μm pore size), and the latter was blocked in TBS–0.05% Tween-20 (TBST)/5% milk and incubated overnight with primary antibodies at 4 °C (Appendix A). After washes in TBST, HRP-conjugated secondary antibodies were incubated in TBST/0.5% milk for 1 h. Immunoreactive bands were detected using Super Signal Chemiluminescent Substrate (Thermo Scientific), and images acquired using Fusion Solo S (Vilber Lourmat, Collégien, Germany).

### 2.10. Statistical Analysis

All graphs and statistical analyses were performed with Prism software (GraphPad Software, San Diego, CA, USA) and are expressed as the mean ± standard deviation. Real-time qPCR values were obtained using the ΔΔCT method [35]. Each graph represents the results obtained from a minimum of four different mice. Nonparametric statistical tests were used: Kruskal–Wallis followed by Dunn’s post-test for multiple comparisons and Mann–Whitney for double comparisons. Differences were considered statistically significant at * *p* < 0.05, ^#^ *p* < 0.01, and ^$^ *p* < 0.001.

## 3. Results

### 3.1. Induction of BRAF^V600E^ Expression in Thyrocytes Triggers a Rapid Increase in Immune Signaling Pathways

Expression of the BRAF^V600E^ oncogene was obtained via intraperitoneal doxycycline injection in the Tg-rtTA/tetO-BRAF^V600E^ mouse model and led to tissue changes similar to those found in human PTC [22]. In our previous work, we observed (i) a twofold increase in thyroid lobe size, (ii) the appearance of papillae in colloidal spaces, (iii) a progressive decrease in the protein reserve contained in the colloid, and (iv) a decrease in the expression of thyrocyte molecular markers, from Day 1 to Day 4 [22]. Additionally, a stromal reaction appeared from Day 2 onward with progressive filling of the interfollicular spaces with cells and development of a dense cellular sleeve around the thyroid lobe by Day 4 (Figure 1A).

In order to understand the TME changes driven by BRAF^V600E^ mutation, we first performed a transcriptomic analysis. Sequencing of total RNA from control (CTL) and BRAF^V600E^ thyroids (Day 2 and Day 4) revealed more than 6000 differentially expressed genes (adjusted *p*-value < 0.05) between Day 2 and CTL thyroids, and more than 9800 differentially expressed genes (adjusted *p*-value < 0.05) between Day 4 and CTL. We eliminated genes with log_2_ fold change between 2 and −2, and the remaining 1269 (Day 2 vs. CTL) and 2716 (Day 4 vs. CTL) genes were analyzed using MouseMine, a pathway enrichment analysis software (Figure 1B,C). After 2 days of BRAF^V600E^ induction, we observed enrichment of several cell cycle- and immune-related signaling pathways (Figure 1B). After 4 days of induction, we found enrichment of pathways associated with homeostasis and a persistence of those linked to the immune system (Figure 1C). Interestingly, immune-related signaling pathways showed the highest number of gene matches (numbers in dark bars of Figure 1C). Among the most deregulated genes on Day 4, we found many chemokines and their respective receptors. We confirmed the sequencing data by measuring the expression of some immune-related genes by RT-qPCR in tissues after 2 and 4 days of doxycycline injections (Figure 1D). Relative expression levels measured by RT-qPCR (blue boxes) were similar to those obtained by sequencing (green dots). Furthermore, we observed an increase in the expression of *CXCR2*, *CCR1*, and *CCR2* receptors, as well as of their ligands *CXCL1*, *CCL7*, and *CCL8*, over the course of doxycycline treatment.

Taken together, these results revealed that induction of BRAF^V600E^ oncogene in thyrocytes caused a rapid stromal response in the thyroid, and that a major aspect of this response is related to immune system signaling pathways.

### 3.2. BRAF^V600E^ Induction Triggers Recruitment of Macrophages, Including a Population of CD11b+/LYVE1+/CD206+/STAB1+ Immunosuppressive Macrophages

Since stromal cellular density increased and since CXCR2, CCR1, and CCR2 receptors are found on the surface of macrophages or myeloid-derived suppressor cells (MDSCs), we studied macrophage recruitment in the thyroid upon BRAF^V600E^ induction. Immunostaining for F4/80 protein, a general macrophage marker, revealed a massive recruitment from Day 2 onward (Figure 2A). On Day 4, each follicle appeared to be closely surrounded by abundant macrophages. Since we observed morphological changes in the thyroid, such as the progressive disappearance of the colloidal content, macrophage recruitment could serve in the cleaning of the tissue. However, lysosomes (LAMP-1+, white) of the F4/80+ macrophages did not contain thyroglobulin (TG, red) on Day 2, and only few of these macrophages exhibited colocalization of LAMP-1 with TG marker, on Day 4 (Appendix A).

We then analyzed by RT-qPCR and in the sequencing data the gene expression of myeloid cell markers, i.e., *Itgam* encoding for CD11b protein, and macrophage-specific genes, i.e., *Adgre1* and *Mrc1* encoding respectively for F4/80 protein and CD206 protein, a marker of immunosuppressive macrophages. Expression of these different genes was increased after BRAF^V600E^ induction (Figure 2B). This increase was due to cell recruitment in the thyroid since the number of CD11b and CD206 immunolabeled cells increased from Day 2 (Appendix A). By RT-qPCR, we also noticed a quantitatively and dynamically similar increase in the expression of *Lyve1* and *Stab1* (Figure 2B). This was interesting since recruitment of CD11b+, F4/80+, LYVE-1+, STABILIN-1+ macrophages has been reported in a melanoma model [36]. Several publications have further demonstrated the presence of LYVE-1, a lymphatic vessel-specific glycoprotein, on macrophages in physiological [37] or pathological [36,38] conditions. STABILIN-1 was first described as a scavenger receptor found on the surface of lymphatic endothelial cells and venous sinusoids [39]. Subsequently, it was also described on the surface of some immunosuppressive macrophages under physiological [40] and pathological [41] conditions.

Co-immunolabeling of LYVE-1, STABILIN-1, and CD206 proteins in CTL thyroids revealed the presence of LYVE-1+ lymphatic structures, but the absence of STABILIN-1 and CD206 positivity (Figure 2C). However, after 2 days of BRAF^V600E^ induction, cells positive for the three markers were detected. The recruitment of those cells increased with time (Day 4) and further specification also occurred since, in addition to triple-positive cells, some cells showed a reduced or undetectable LYVE-1 marker (Figure 2C; marked with an asterisk in bottom panel).

Altogether, these results demonstrated a massive and rapid recruitment of macrophages following BRAF^V600E^ induction. A minority of these macrophages might be involved in endocytosis of proteins or cellular debris. On the contrary, a population of macrophages carrying immunosuppressive markers, such as CD206 and STABILIN-1 appeared in the thyroid 2 days after BRAF^V600E^ induction, and persisted in the tissue.

### 3.3. Generation and Validation of a CRISPR/Cas9 Stabilin-1 Knockout Mouse

Several publications have demonstrated that STABILIN-1 expression on macrophages plays a role as an immunosuppressive protein [42]. In different models of subcutaneous allografts of cancer cells, tumors presented reduced size in the absence of STABLIN-1 [43,44], and decreased metastasis [43,45]. However, the role of STABILIN-1+ macrophages in thyroid cancer or in an in situ tumor model is unknown. We, therefore, took advantage of our in situ PTC model to study the role of these macrophages by combining CRISPR/Cas9 knockout of *Stabilin-1* with Tg-rtTA/tetO-BRAF^V600E^ transgenes.

We designed two guide RNAs to ablate the transcription initiation site, as well as the entire first exon, containing the ATG start codon, out of the 69 exons of the *Stab1* gene (Figure 3A, scissors). These allowed removal of a DNA fragment of approximately 1300 bp, as verified by PCR (Figure 3B). Amplicons generated with primers located outside and inside of the targeted regions (fw–rv–rv’), had a band size of ±2300 bp and ±1300 bp for the wildtype allele and ±1000 bp for the deleted allele (Figure 3B, left). Amplification of the longest (±2300 bp) product was highly variable (here, only visible in heterozygous (+/−) mice). A second forward primer (fw′), located within the first exon, was also used with the reverse primer (rv’) (Figure 3B, right). Amplification of a ±600 bp band revealed the presence of at least one wild-type allele (in +/+ and +/−), while no amplification was possible in knockout mice (−/−) since both primers (fw′-rv’) were in the deleted region. *Stabilin-1* knockout embryos were obtained at the expected Mendelian ratio, and they were indistinguishable from control littermates at adult age. Western blotting analyses revealed the presence of a 260 kDa band in wildtype (+/+) adult lymph nodes, which was undetectable in knockout (−/−) organs (Figure 3C). Since we inactivated a gene expressed by a cellular population of the immune system, we analyzed the most important immune populations in the blood, spleen, and thymus of adult unchallenged mice by flow cytometry (Figure 3D–F). The percentage of the different immune populations was comparable in wildtype (*Stab1*^+/+^) and KO mice (*Stab1*^−/−^). The CRISPR/Cas9 *Stab1* inactivation, thus, led to a complete absence of STABILIN-1 full-length protein with no detectable changes in immune populations.

### 3.4. Absence of STABILIN-1 Does Not Affect Epithelial PTC Development in Mice

To investigate if the absence of STABILIN-1 could impact on PTC development, wildtype (*Stab1*^+/+^) and KO (*Stab1*^−/−^) mice were treated with doxycycline every 24 h for 4 days. Tissues were analyzed on Day 2 and Day 4. Hematoxylin–eosin staining (Figure 4A) revealed similar morphological changes following BRAF^V600E^ induction in *Stab1*^+/+^ and *Stab1*^−/−^ mice, i.e., (i) increase in lobe size, (ii) progressive loss of colloid protein content, and (iii) appearance of papillae. Activity of the MAPK pathway, constitutively activated by the BRAF^V600E^ mutation, was assessed by measuring the expression of two target genes *Fosl1* and *Dusp5* by RT-qPCR (Figure 4B). Regardless of mouse genotype, expression of these two genes was strongly increased on Day 2, and slightly less on Day 4. Immunohistochemistry of phospho-ERK protein in the *Stab1*^+/+^ and *Stab1*^−/−^ thyroid lobes also confirmed the similar activation of the MAPK pathway as early on Day 2 and its maintenance on Day 4 (Appendix A). Lastly, we measured by RT-qPCR the expression of different genes involved in thyroid function, *Nis*, *Tpo*, *Tg*, and *Tshr*, as well as of two thyroid transcription factors, *Nkx2.1* and *Pax8* (Figure 4C). For *Nis*, *Tpo*, *Tg*, and *Tshr*, the decrease was progressive. On Day 2 and Day 4, expression of all these thyroid genes was decreased in a comparable manner in *Stab1*^+/+^ and *Stab1*^−/−^ mice. These results indicated that, within 4 days, the absence of STABILIN-1 did not impact on epithelial characteristics of BRAF^V600E^-induced PTC development.

### 3.5. Absence of STABILIN-1 Does Not Affect Circulating Cell Populations or Macrophage Recruitment in BRAF^V600E^-Induced PTC

To assess circulating cell populations, we performed blood tests on *Stab1*^+/+^ and *Stab1*^−/−^ mice in CTL condition, on Day 2 and Day 4 after BRAF^V600E^ induction. The numbers of white blood cells (WBC) and platelets were around 7 × 10^3^ and 700 × 10^3^ per mm^3^ of blood, respectively, in CTL (PBS-injected) *Stab1*^+/+^ mice. These numbers were not affected by BRAF^V600E^ induction on Day 2 and Day 4, nor by the absence of STABILIN-1 (*Stab1*^−/−^) (Appendix A). Similarly, the percentage of the different circulating cell populations was not affected by BRAF^V600E^ induction (Day 2 and Day 4 vs. CTL), nor by the genotype (*Stab1*^−/−^ vs. *Stab1*^+/+^), with ±75% of lymphocytes, ±3% of monocytes, ±12% of granulocytes, and ±7% of eosinophils (Appendix A). Lastly, we measured and characterized, by flow cytometry, more specific circulating cell populations after 4 days of BRAF^V600E^ induction. After gating on live and CD45+ cells, we separated CD4+, CD8+ T cells, natural killer cells (CD19− CD49b+), B cells (CD4−, CD8−, CD19+), and myeloid cells (CD11b+) (Appendix A). Again, no change in abundance of these populations was observed in *Stab1*^−/−^ mice, as compared with *Stab1*^+/+^ mice.

We then studied macrophage recruitment in thyroid tissue 4 days after BRAF^V600E^ expression in *Stab1*^+/+^ and *Stab1*^−/−^ mice. We first measured by RT-qPCR the expression of the different macrophage markers analyzed previously, *Itgam*, *Adgre1*, *Mrc1*, and *Lyve1* (Figure 5A). Expression of these genes was comparable in both *Stab1* genotypes. Conversely, using the same material, we observed in the *Stab1*^−/−^ mice a strong decrease (about 10-fold) in *Stab1* gene expression, not compensated for by an increase in the expression of *Stabilin-2* gene (*Stab2*) (Figure 5A). Please note that, despite a 10-fold decrease in *Stab1* gene expression, we did not observe a complete absence of transcript. This can be explained by the use of primers in exons 49 and 51 and an illegitimate transcription of the *Stab1* gene at a low rate, despite deletion of the transcription initiation site in the knockout allele (Figure 3A). We then confirmed the presence of macrophages in the thyroid by immunolabeling (Figure 5B). In line with the RT-qPCR results, CD206+ cells were observed around the follicles in *Stab1*^+/+^, as well as in *Stab1*^−/−^ mice, i.e., in the absence of STABILIN-1 staining.

Lastly, we evaluated the expression profile of different markers and cytokines of proinflammatory (*TNFα*, *Nos2*, *Il1b*, and *Il6*) and anti-inflammatory (*Mrc1*, *Arg*, and *Il10*) macrophages in CTL, Day 2, and Day 4 thyroid tissue in *Stab1*^+/+^ and *Stab1*^−/−^ mice (Appendix A). We found a rapid increase in the expression of proinflammatory cytokines (*TNFα*, *Il1b*, and *Il6*) 2 days after BRAF^V600E^ induction. Conversely, there was a slight delay in the response for anti-inflammatory macrophage markers and cytokines (*Mrc1*, *Arg* and *Il10*) which only displayed a significant increase on Day 4. These changes in gene expression were independent of the *Stab1* genotype (Appendix A). In a subcutaneous allograft model, Viitala et al. showed that the plasma concentration of TNFα was increased in *Stab1*^−/−^ mice [44]. We, thus, measured TNFα concentration in *Stab1*^+/+^ and *Stab1*^−/−^ mice, 4 days after in situ BRAF^V600E^-dependent PTC development. We also observed a trend toward increased plasma TNFα concentration in the in situ PTC mouse model, but without reaching significance. Altogether, our data indicate that the absence of STABILIN-1 did not affect macrophage recruitment triggered by BRAF^V600E^ expression but the increased trend in plasma TNFα concentration might suggest an increased proinflammatory response in the absence of STABILIN-1.

### 3.6. Absence of STABILIN-1 Changes the CD4+/CD8+ T-Cell Ratio in BRAF^V600E^-Dependent PTC

It was demonstrated that blocking STABILIN-1 on macrophages or inactivating *Stab1* allowed the reactivation of CD8+ T cells within the tumor [44]. Thus, we first dissociated *Stab1*^+/+^ and *Stab1*^−/−^ thyroid tissue 4 days after BRAF^V600E^ induction, and then quantified the percentages of CD4+ and CD8+ T cells from live/CD45+/CD19−/CD11b−/CD49b−DX5− cells (Appendix A). Out of four independent experiments, the ratio of CD4+ cells to CD8+ cells tended to decrease in *Stab1*^−/−^ mice compared to *Stab1*^+/+^ mice (Appendix A). However, this was not significant. Since these experiments required substantial biological material (eight lobes/condition/experiment), we turned to immunohistofluorescence (Figure 5D,E). We used two lymphocyte markers, CD3 (green) and CD8 (red), in combination with epithelial cadherin (E-CADHERIN, white) to reveal the thyroid epithelium (Figure 5E). After nuclei segmentation, the number of each cell type was counted with the HALO software in CTL, Day 2, and Day 4 thyroid tissue in *Stab1*^+/+^ and *Stab1*^−/−^ mice. Due to the lack of a validated CD4 antibody working on sections, CD3+ CD8− cells were considered as CD4+ T cells. CD3+ CD8+ T cells were considered as CD8+ T cells. CD4+ T cells were present in the thyroid before induction of BRAF^V600E^ expression, and their number increased with time. CD8+ T cells were absent in CTL thyroids and were recruited during PTC development (Appendix A). On the basis of these quantifications, we calculated the CD4+/CD8+ ratio. On Day 2, the ratio of CD4+ T cells to CD8+ T cells was slightly decreased in *Stab1*^−/−^ mice (Figure 5D). This trend was confirmed on Day 4 when the ratio dropped from ±40 to ±20 in *Stab1*^−/−^ mice. This twofold decrease was due to a slight decrease in the percentage of CD4+ cells and a slight increase in CD8+ cells in *Stab1*^−/−^ mice, as compared to *Stab1*^+/+^ mice (Appendix A).

Thus, the absence of STABILIN-1 caused a change in the lymphocytic populations present in the thyroid tissue, resulting in a decreased CD4+/CD8+ T-cell ratio, supporting a potential immunosuppressive role for STABILIN-1 expression by macrophages recruited in the PTC thyroid tissue.

### 3.7. STABILIN-1+/CD206+ Cells Are Present in Benign and Malignant Human Thyroid Tissues

To extend our mouse data to patients diagnosed with thyroid diseases and in particular cancer, we investigated the presence of STABILIN-1+/CD206+ cells in benign tumors (multinodular goiter, MNG, *n* = 3), in healthy tissue neighboring a PTC (NHT, *n* = 6) and in PTC (*n* = 12) (Figure 6A). Interestingly, these STABILIN-1+/CD206+ macrophages were present and predominantly localized in the peripheral areas of the tumors for MNGs and PTCs. In MNG tissues, it seemed that the majority of STABILIN-1+ cells were not CD206+, while, in PTCs, we found different populations: STABILIN-1+ cells, CD206+ cells, and CD206+/STABILIN-1+ cells (Figure 6A, tissue periphery). Within the tumor core, i.e., where E-CADHERIN density was very high, we found very few CD206+ and/or STABILIN-1+ cells (Figure 6A, tissue center). Lastly, in the neighboring healthy tissues (NHT), we found STABILIN-1+ cells with or without CD206 co-staining either within the epithelial thyroid tissue or at the periphery.

Lastly, we measured by RT-qPCR the expression of different gene markers for immune cell populations in thyroid pathologies, such as autoimmune diseases (Basedow, *n* = 11 and Hashimoto, *n* = 4), as well as benign (MNG, *n* = 8) or malignant tumors (PTC, *n* = 7) (Figure 6B). We found no statistical differences in the expression of genes related to immunosuppressive macrophages (*STAB1* and *MRC1*), myeloid cells (*ITGAM*), or T cells (*CD4* and *CD8*) in these pathological conditions. Of note, samples from Hashimoto’s patients seemed to display a higher expression for these different genes, correlating with their description as chronic lymphocytic thyroiditis [46]. This preliminary analysis revealed the presence of STABILIN-1+ cells in human diseased thyroid parenchyma, without showing a specific expression pattern.

## 4. Discussion

Although TC generally presents a 5 year overall survival rate higher than 95%, some tumors remain very aggressive and difficult to cure [6]. Among solid tumors with poor prognosis, it seems that, in addition to tumor cells, the TME and the cells which compose this environment play a very important role in tumor progression [9]. By using a validated genetically engineered, doxycycline-inducible mouse model of BRAF^V600E^-driven papillary thyroid cancer that mimics human PTC, we aimed to elucidate the changes in the tumor microenvironment induced by the BRAF^V600E^ mutation. Starting from a transcriptomic analysis, we identified major changes in immune signaling pathways associated with recruitment of a particular population of LYVE-1+/CD206+/STABILIN-1+ macrophages in the thyroid parenchyma. CRISPR/Cas9 inactivation of the *Stabilin-1* gene in the in situ PTC model revealed an increase in the number of CD8+ T cells in the thyroid TME during tumor progression. Lastly, we also demonstrated the presence of CD206+/STABILIN-1+ cells in the parenchyma of benign and malignant human thyroid tumors.

The PTC model chosen for this study is a very powerful model. As soon as doxycycline is administered to mice, all thyroid cells start to express the BRAF oncogene. Moreover, the expression of the oncogene does not depend on its endogenous promoter but on a bacterial promoter, which causes strong and sustained transcriptional activation. Despite this non-physiological BRAF^V600E^ expression, this model has already made it possible to study MAPK pathway inhibitors that could be used as treatment [21]. Ryder et al. also highlighted the necessity of the CCR2 receptor for macrophage attraction and tumor development using this PTC model [19]. This PTC mouse model indeed presents several advantages: (i) ease of use as it only depends on the presence of two transgenes and the administration of doxycycline [47]; (ii) flexibility in the timing of oncogene induction and in the dose of doxycycline administered (either via food or via intraperitoneal injections) [21,22]; (iii) short latency and high reproducibility as compared to sporadic thyroid cancer models [48,49,50]; (iv) possibility to study early changes occurring in the thyroid which allows to investigate in situ PTC development as well as the thyroid immune microenvironment, impossible to study in xenograft models in immunosuppressed mice. Indeed, recruitment of TAMs described by Ryder et al. in human PTCs is correctly reproduced in this PTC mouse model [51]. Our RNA-seq data also support the comparative analysis of the TCGA and ESTIMATE databases performed by Zhao et al. which shows that the hub-regulated genes are mainly related to immune signaling pathways with a majority of chemokine and cytokine genes identified [52].

The present study highlighted the recruitment of a particular population of immunosuppressive macrophages carrying LYVE-1/CD206/STABILIN-1 markers. This population has already been described in different types of tumor cell allografts from different organs (melanoma, lung carcinoma, lymphoma, and colon cancer) [36,43,44]. In the melanoma allograft model, STABILIN-1 expression is induced on monocytes and on the tumor vasculature [43]. In the absence of STABILIN-1 on macrophages or vessels, tumor size is smaller, and the number of metastases is lower. These changes are accompanied by a reduction in the number of CD206+ macrophages found within the tumor. In the lung carcinoma allograft model, tumors are also smaller in the absence of STABILIN-1, and this reduction in size is even more surprising if *Stabilin-1* is inactivated only on macrophages [44]. In this case, a decrease in the number of TAMs is observed but the TAMs present in the tumor express more MHC-II and CD206. In the PTC model used in this study, we did not observe STABILIN-1 expression on tumor vessels or changes in CD206+ macrophage populations upon *Stabilin-1* inactivation. This could have resulted from the short readout time (only 4 days). Indeed, absence of STABILIN-1 visualization on tumor vessels could be explained by the slower induction of STABILIN-1 expression on vessels as compared to TAMs, as described in [44]. Alternatively, STABILIN-1 immunolocalization could be not sensitive enough to visualize small changes in protein expression on vessels. It would, therefore, be interesting to monitor *Stabilin-1* expression by in situ hybridization coupled with immunohistofluorescence to visualize the populations of interest [53,54].

The lack of tumor size reduction in the absence of STABILIN-1 could be due to a compensation by STABILIN-2. Indeed, STABILIN-1 and STABILIN-2 are 55% identical at the protein level [41]. However, STABILIN-2 has never been described as a putative rescuer protein, and its expression was not changed in *Stab1^−/−^* (Figure 5A). Another explanation for the lack of tumor size reduction in the absence of STABILIN-1 could also be the short readout time of the model used. Indeed, we observed a tendency to decreased CD4+/CD8+ T-cell ratio on Day 2 and a significant decline after 4 days of doxycycline treatment. If the PTC mouse model could be analyzed at later timepoints, which is problematic, we would expect to observe a reduction in tumor size. However, we should not forget that STABILIN-1 expression on lymphatic vessels has been shown to play an important role in lymphocyte trafficking to the draining lymph nodes and in dendritic cell entrance from the site of inflammation to the lymph nodes [55,56]. Thus, the total absence of STABILIN-1 might impact trafficking of these immune cells to and from the PTC, e.g., the CD8+ T cells, which are mainly responsible for the decreased tumor size. We did not take the advantage of the macrophage-specific *Stabilin-1* knockout model (as in the study of Viitala et al. [44]), since it would have been complicated and time-consuming to implement in the in situ PTC model, already depending on the presence of two different transgenes

Mechanistically, the role of STABILIN-1 macrophages in BRAF^V600E^-dependent in situ PTC is consistent with immunosuppression via their interaction with T cells and the change in the TNFα plasma concentration, as described in the different allograft models [44]. STABILIN-1 has also been proposed to be a scavenger receptor important for the clearance of extracellular tumor growth-inhibiting molecules, such as SPARC or chitinase-like protein, from the TME in some tumor types (breast cancer and neuroblastoma), thereby leading to tumor growth [57,58,59]. Since these two STABILIN-1 ligands are rather pro-tumoral in human PTC [60,61], their clearance by STABILIN-1 TAMs could therefore be detrimental for thyroid tumor growth.

In human, the expression level of STABILIN-1 correlates with immunosuppressive environments such as the placenta or tumors [40]. In preeclampsia, STABILIN-1 expression on placental macrophages and circulating monocytes is decreased [62]. In different cancer (breast, head and neck, and colorectal), the presence of STABILIN-1 on tumor lymphatic vessels facilitates metastatic spreading to lymph nodes, and its presence on intratumoral or peritumoral macrophages is associated with a poor prognosis [63,64]. The expression of STABILIN-1 on TAMs in urothelial bladder cancer is used as a biomarker since its presence is associated with increased mortality and poor response to chemotherapy [65,66]. The risk of recurrence of oral cavity cancer correlates with the density of STABILIN-1 TAMs [67]. Lastly, patients treated by immunotherapy with checkpoint inhibitors have a reduced response to the treatment if they express a high level of STABILIN-1 [42]. It is, therefore, important to consider STABILIN-1 expression level in tumors to adjust the treatment. To the best of our knowledge, no study has described the presence of STABILIN-1 cells in thyroid pathologies. Our preliminary analysis revealed its expression on TAMs in benign and malignant tumors but a larger number of samples, as well as additional analyses, are necessary to consider a role for STABILIN-1 macrophages in thyroid pathologies. Future work should investigate the level of expression of STABILIN-1 in different populations, such as vascular endothelial cells, lymphatic endothelial cells, and myeloid cells, present in the microenvironment of autoimmune or cancerous pathologies. It would also be relevant to analyze the correlation between *STABILIN-1* expression and the genetic status of the tumor such as the BRAF^V600E^ mutation, frequently found in PTCs and in melanomas where the presence of STABILIN-1 TAMs has also been described [36].

## 5. Conclusions

Our work demonstrated rapid immune changes occurring in the thyroid parenchyma following expression of the BRAF^V600E^ oncogene in thyrocytes. This implicates the recruitment of a particular population of STABILIN-1+ immunosuppressive macrophages never described before in PTC. The absence of this protein changes the immune status of the tumor with the recruitment of CD8+ T cells into the tumor. It would, thus, seem that, as in allograft models, STABILIN-1+ TAMs play an immunosuppressive role in in situ PTC progression.

## Figures and Tables

**Figure 1 cancers-14-04687-f001:**
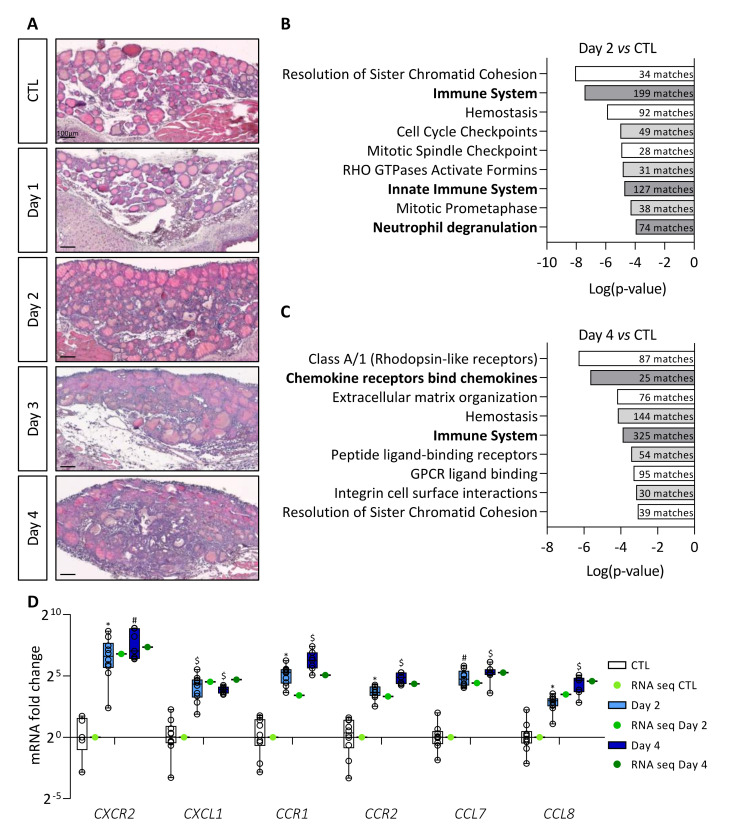
BRAF^V600E^ induction in mouse thyrocytes triggers stromal reaction which correlates with major changes in immune signaling pathways. (**A**) Hematoxylin–eosin staining of thyroid tissue from control (CTL) and doxycycline-treated mice, from Day 1 to Day 4. (**B**,**C**) Enriched signaling pathway revealed by MouseMine software using differentially expressed genes from total RNA sequencing (**B**) of Day 2 (*n* = 4) compared to CTL (*n* = 4) thyroids and (**C**) of Day 4 (*n* = 4) compared to CTL (*n* = 4) thyroids. (**D**) mRNA fold change of the most deregulated chemokines (*CXCL1*, *CCL7*, *CCL8*) and their receptors (*CXCR2*, *CCR1*, *CCR2*) found in RNA sequencing data (green dots) and measured by RT-qPCR (white and blue boxes) of CTL (*n* ≥ 5) and doxycycline-treated mice, after 2 (*n* = 11) and 4 days (*n* = 7) of injections (Day 2, Day 4). mRNA fold changes are normalized to the geometric mean of *Rpl27* and *Gapdh* expression and compared to control group. Data are expressed as the mean ± SD (* *p* < 0.05, ^#^ *p* < 0.01, ^$^ *p* < 0.001).

**Figure 2 cancers-14-04687-f002:**
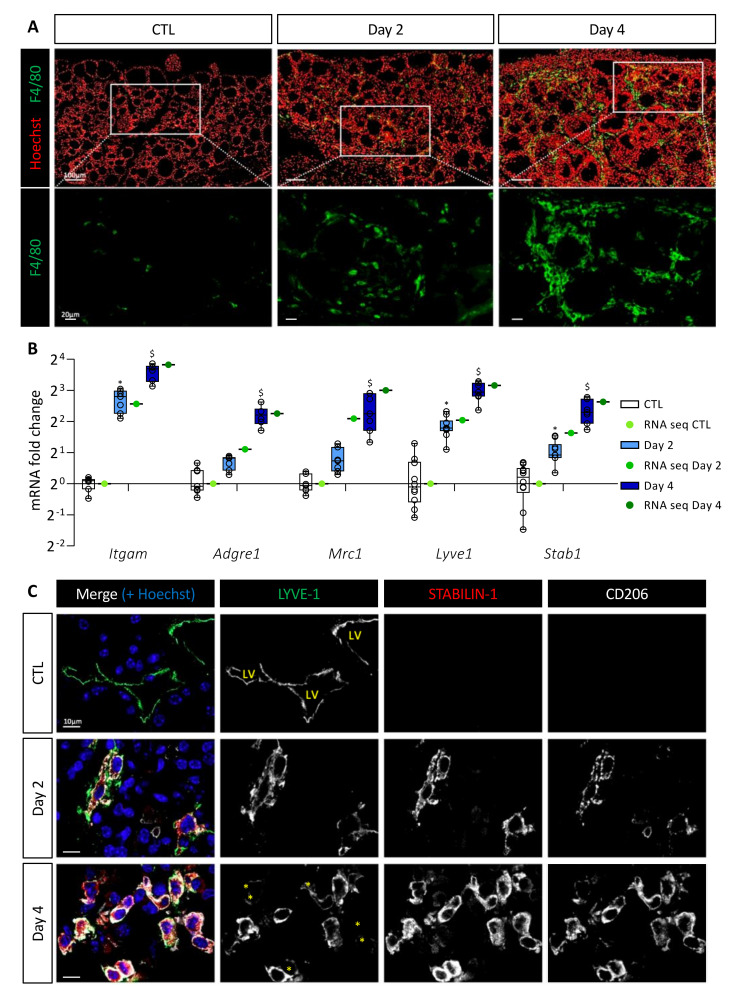
BRAF^V600E^ expression in mouse thyrocytes leads to massive macrophage recruitment including a particular macrophage population characterized by LYVE-1, CD206, and STABILIN-1 expression. (**A**) Immunohistofluorescence of F4/80 (green) on thyroid tissues from control (CTL) and doxycycline-treated mice (Day 2, Day 4). Hoechst (red) is used to label the nuclei. (**B**) mRNA fold change of different macrophage markers found in RNA sequencing data (green dots) and measured by RT-qPCR (white and blue boxes) from CTL (*n* = 10) and doxycycline-treated mice, after 2 (*n* = 11) and 4 days (*n* = 7) of injections (Day 2, Day 4). mRNA fold changes are normalized to the geometric mean of *Rpl27* and *Gapdh* expression and compared to control group. Data are expressed as the mean ± SD (* *p* < 0.05, ^$^ *p* < 0.001). (**C**) Immunohistofluorescence of LYVE-1 (green), STABILIN-1 (red), and CD206 (white) on thyroids from control (CTL) and doxycycline-treated mice (Day 2, Day 4). Hoechst (blue) is used to label the nuclei. LV indicates the lumen of lymphatic vessels, and yellow asterisks denote cells with a very weak signal for LYVE-1 but positive for STABILIN-1 and CD206 markers.

**Figure 3 cancers-14-04687-f003:**
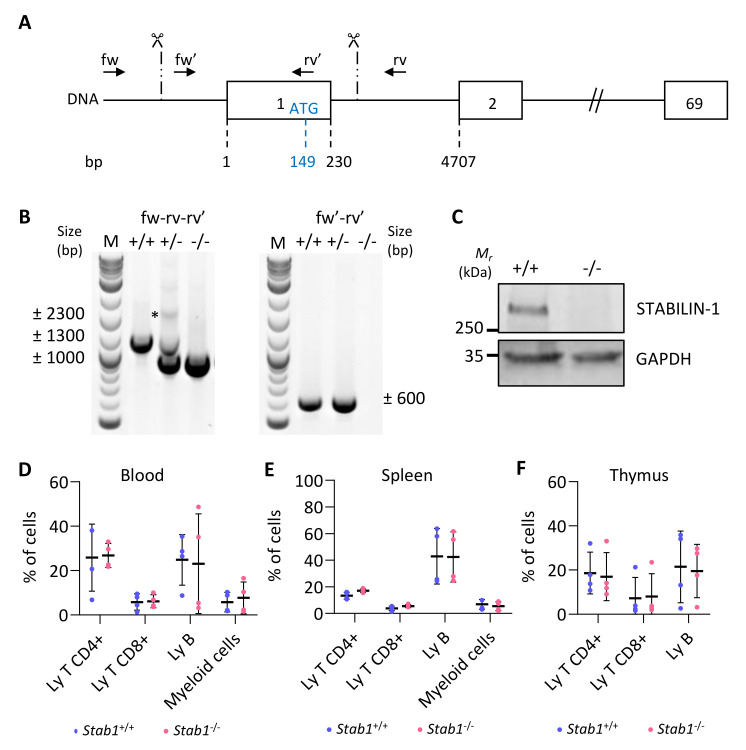
CRISPR/Cas9-based editing of *Stabilin-1* gene does not affect immune status of unchallenged mice. (**A**) Schematic representation of the *Stab1* locus with exons represented as boxes (only exons 1, 2, and 69 out of the 69 are illustrated). Numbers below indicate the position of the exon limits with respect to the transcription initiation site (+1). The position of the ATG (149 bases downstream of the +1) in the first exon is also indicated. Scissors and vertical dashed lines indicate the approximate region where the cuts occurred. Localization of genotyping primers is represented by black arrows. (**B**) Illustrative genotyping results for wildtype (+/+), heterozygous (+/−), and knockout (−/−) mice, obtained with two sets of primers. *, denote the absence of the theoretical wildtype band in +/+ mice. (**C**) Western blotting of STABILIN-1 protein in lymph node lysate from wildtype (+/+) and knockout (−/−) mice. GAPDH was used as a loading control. (**D**–**F**) Flow cytometry analyses of CD4+ T cells (CD45+, CD11b−, CD19−, CD3+, CD4+, CD8−), CD8+ T cells (CD45+, CD11b−, CD19−, CD3+, CD4−, CD8+), B cells (CD45+, CD11b−, CD19+), and myeloid cells (CD45+, CD11b+) in blood (D) and from dissociated cells of spleen (**E**), and thymus (**F**) from wildtype (+/+) and knockout (−/−) mice (*n* = 4).

**Figure 4 cancers-14-04687-f004:**
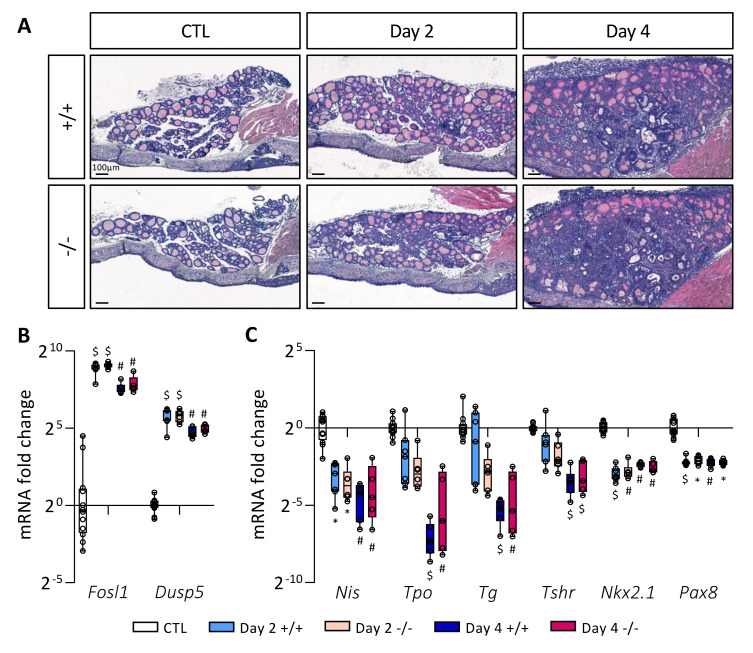
Absence of STABILIN-1 does not affect PTC development triggered by BRAF^V600E^ expression. (**A**) Hematoxylin–eosin staining of thyroid tissue from wildtype (+/+) and knockout (−/−) mice treated with doxycycline (Day 2 and Day 4) or not (CTL). (**B**,**C**) RT-qPCR analyses of two MAPK pathway-target genes (B) and thyroid markers (**C**) in thyroid tissues from wildtype (+/+) and knockout (−/−) mice treated with doxycycline (Day 2 and Day 4) or not (CTL) (*n* ≥ 5). mRNA fold changes are normalized on the geometric mean of *Rpl27* and *Gapdh* expression and compared to the CTL group (seven +/+ and five −/− thyroid samples). Data are expressed as the mean ± SD (* *p* < 0.05, ^#^ *p* < 0.01, ^$^ *p* < 0.001).

**Figure 5 cancers-14-04687-f005:**
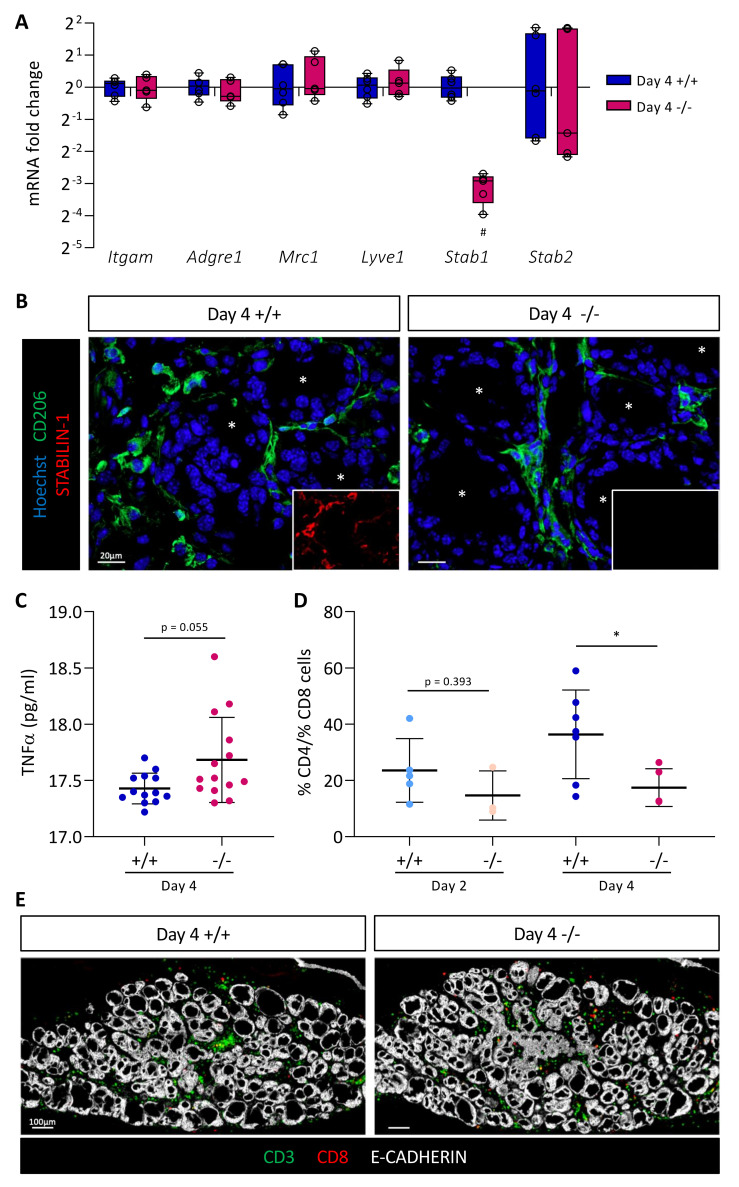
Absence of STABILIN-1 does not affect macrophage recruitment during PTC induction but changes the ratio of CD4+ T cells to CD8+ T cells. (**A**) RT-qPCR analyses of macrophages markers and *Stabilin-2* (*Stab2*) genes in thyroid tissues from wildtype (+/+) and knockout (−/−) mice treated with doxycycline for 4 days (*n* ≥ 5). mRNA fold changes are normalized to the geometric mean of *Rpl27* and *Gapdh* expression and compared to wildtype group (+/+). Data are expressed as the mean ± SD (^#^ *p* < 0.01). (**B**) Immunohistofluorescence of CD206 (green) and STABILIN-1 (red, in small boxes) on thyroid tissue sections from wildtype (+/+) and knockout (−/−) mice treated with doxycycline for 4 days (Day 4). Hoechst (blue) was used to label the nuclei. White asterisks denote the lumen of thyroid follicles. (**C**) Plasmatic TNFα concentration (pg/mL) measured by ELISA from wildtype (+/+) and knockout (−/−) mice treated with doxycycline for 4 days (Day 4). (**D**) Ratio of CD4+ T cells to CD8+ T cells calculated after the quantifications of immunohistofluorescently labeled CD3 and CD8 cells in thyroid tissue sections from wildtype (+/+) and knockout (−/−) mice treated with doxycycline (Day 2 and Day 4). CD4+ T cells were considered as CD3+ CD8- cells. Each point represents the mean number of cells quantified from minimum three different (nonadjacent) sections of the same lobe (*n* ≥ 4, * *p* < 0.05). (**E**) Illustrative immunofluorescence, used for cell quantifications, of CD3 (green), CD8 (red), and the epithelial marker (E-CADHERIN, white) on thyroid tissue sections from wildtype (+/+) and knockout (−/−) mice treated with doxycycline for 4 days (Day 4).

**Figure 6 cancers-14-04687-f006:**
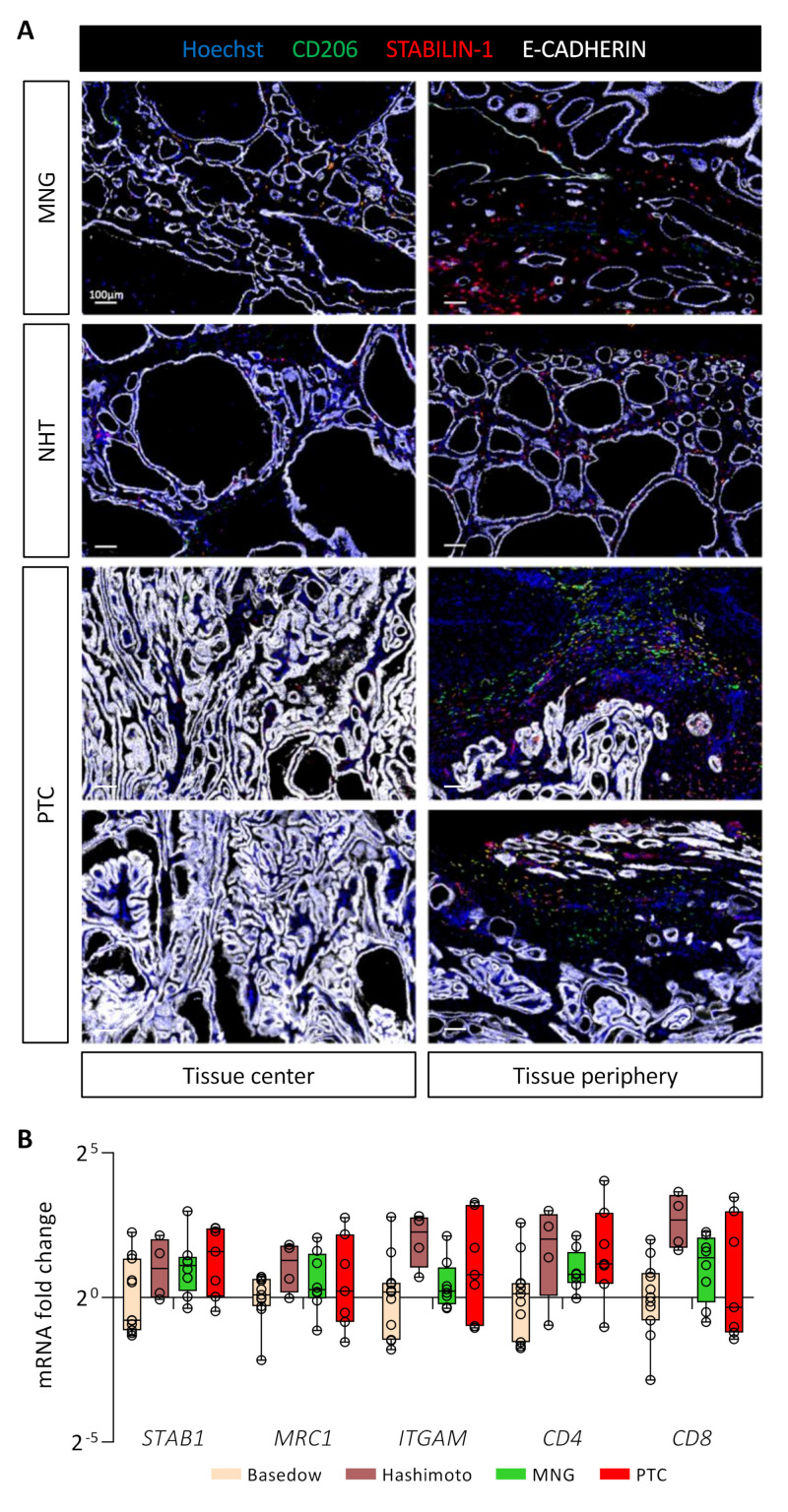
STABILIN-1+ cells are found in benign and malignant human thyroid tumors. (**A**) Immunohistofluorescence of CD206 (green), STABILIN-1 (red), and E-CADHERIN (white) on sections of human thyroid conditions: multinodular goiter (MNG), healthy tissue neighboring PTC (NHT), and papillary thyroid carcinoma (PTC). For each condition, a picture in the center and the periphery of the tissue is shown. (**B**) RT-qPCR analyses of different immune cell population marker genes on different thyroid pathologies. mRNA fold changes are normalized to the *RPL13A* expression and compared to Basedow disease. Data are expressed as the mean ± SD.

## Data Availability

RNA sequencing data are available by following this link: https://www.ncbi.nlm.nih.gov/bioproject/878663 (accessed on 20 September 2022).

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
