# Peer review of "BRAFV600E Expression in Thyrocytes Causes Recruitment of Immunosuppressive STABILIN-1 Macrophages"

_cancers, 2022, doi:10.3390/cancers14194687_

Round 1
Reviewer 1 Report
The authors Spourquet et al. describe very well changes in immune signaling pathways induced by BRAFV600E mutation.
Nevertheless, I see some open questions for the reader and in some places I miss essential details that would be desirable for understanding and reconstructing the results presented.
- missing information about total RNA-Seq: #reads sequenced, strategie, library, etc
- seq. data should be publically available, also table of DE analysis should be provided to the reader
- how was the sequencing raw data analyzed in detail, quality, mapping, annotation, genome and annotation version, which tools and parameters other from default used?
- how was DE performed, tools, parameters other from default
- MouseMine parameter
- in general, all tools should be mentioned as well as parameters used
- can mouse model be validated in some way, by RNA-seq, Immunohistofluorescence, etc, .. cannot see the expression changes of BRAF in the presented data
- where BRAF V600E mutation checked in RNA-Seq? is the mutation visible in all samples? would be good to know, see BRAF expression above
- what is non of leakyness of the system, anything known by the authors?
- could the reader be provided a schematic representation of the mouse model transgene?
Reviewer 2 Report
Papillary thyroid carcinoma (PTC) is a histological subtype of thyroid cancer that carries BRAFV600E, an activating mutation. BRAFV600E is associated with malignant transformation, therapy resistance, and tumor progression. Spourquet and collaborators used a doxycycline-inducible model of BRAFV600E that only expresses BRAFV600E in thyrocytes to study the molecular determinants of PTC. The authors used histological, transcriptomics, and immunofluorescence assays and described a STABILIN-1 population of macrophages recruited to the neoplasm. The following statements should be clarified.
-RT qPCR methodology must be by the MIQE guidelines
Stephen A Bustin, Vladimir Benes, Jeremy A Garson, Jan Hellemans, Jim Huggett, Mikael Kubista, Reinhold Mueller, Tania Nolan, Michael W Pfaffl, Gregory L Shipley, Jo Vandesompele, Carl T Wittwer, The MIQE Guidelines: Minimum Information for Publication of Quantitative Real-Time PCR Experiments, Clinical Chemistry, Volume 55, Issue 4, Apr 1 2009, Pages 611–622, https://doi.org/10.1373/clinchem.2008.112797
In the Western Blotting section, specify the pore of the PVDF membrane.
In line 476 authors state, “Altogether, our data indicate that the absence 476 of STABILIN-1 did not affect macrophage recruitment triggered by BRAFV600E expression, but the increased trend in plasma TNF concentration might suggest an increased pro-inflammatory response in the absence of STABILIN-1”. Nevertheless, the authors shouldn't imply that the trend in TNF might suggest an increased pro-inflammatory response. Authors must present data from other pro-inflammatory cytokines with statistical differences, and from the mRNA sequencing analysis, please describe the expression of pro and anti-inflammatory cytokines.
In line 69, the authors described that TC incidence is higher in women than in men. In this regard, is data sex-paired? If that is the case, please specify; if not, please do the statistical analysis considering sex. Authors should consider that the inflammatory response shows sexual dimorphism in mice:
Villapol S, Loane DJ, Burns MP. Sexual dimorphism in the inflammatory response to traumatic brain injury. Glia. 2017 Sep;65(9):1423-1438. DOI: 10.1002/glia.23171. Epub 2017 Jun 13. PMID: 28608978; PMCID: PMC5609840.
Du XJ. Gender modulates cardiac phenotype development in genetically modified mice. Cardiovasc Res. 2004 Aug 15;63(3):510-9. DOI: 10.1016/j.cardiores.2004.03.027. PMID: 15276476.
The authors should discuss in detail if another member of the STABILIN family was compensating for STABILIN-1 deletion.
Reviewer 4 Report
The article is up to date and following the new era.
Abstract: Line 43 is unclear to the reader. Immunosuppressive in which domain. Line 125 "ratio" may be added.
Introduction: lines 115-117 needs reference.
Methodology: Why these particular genes Line 207 were selected. Can you support by reference?
Results: I ran single cell RNAseq analysis on human samples to have a more confirmation for STAB1 gene expression at the cellular level. The gene was mostly expressed in 78% to 99% of macrophages, T lymphocytes, and epithelium in primary tumors and nodal and subcutaneous metastasis (GSE184362).
Line 310: Is there a reference or notice of authors?
Suggest checking the final gene panel in TCGA and cell lines to explain the role of the final selected genes.
Round 2
Reviewer 2 Report
Spourquet and collaborators used a doxycycline-inducible model of BRAFV600E to study the molecular determinants of PTC. They improved their manuscript according to the suggestions.
The methodology section is more straightforward.
The discussion is well-structured.